# Hyperbaric oxygen therapy for radiation-induced tissue injury following sarcoma treatment: A retrospective analysis of a Dutch cohort

Jasmijn D. Generaal[1], Corine A. Lansdorp[2], Onno Boonstra[2], Barbara L. van Leeuwen[1], Hubertus A. M. Vanhauten[3], Marc G. Stevenson[1], Lukas B. Been[1]*

1 Division of Surgical Oncology, Department of Surgery, University Medical Center Groningen, University of Groningen, Groningen, The Netherlands, 2 Institute for Hyperbaric Medicine, Rotterdam, The Netherlands, 3 Department of Radiation Oncology, University Medical Center Groningen, University of Groningen, Groningen, The Netherlands

* l.b.been@umcg.nl

## Abstract

### Background and objectives

Sarcomas are commonly managed by surgical resection combined with radiotherapy. Sarcoma treatment is frequently complicated by chronic wounds and late radiation tissue injury (LRTI). This study aims to gain insight in the use and results of hyperbaric oxygen therapy (HBOT) for radiation-induced complications following sarcoma treatment.

### Methods

All sarcoma patients treated between 2006 and 2017 in one of the five centers of the Institute for Hyperbaric Oxygen in the Netherlands were included for retrospective analysis.

### Results

Thirty patients were included, 18 (60.0%) patients were treated for chronic wounds and 12 (40.0%) for LRTI. Two patients with chronic wounds were excluded from analysis as HBOT was discontinued within five sessions. In 11 of 16 (68.8%) patients treated for chronic wounds, improved wound healing was seen. Nine of 12 (75.0%) patients treated for LRTI reported a decline in pain. Reduction of fibrosis was seen in five of eight patients (62.5%) treated for LRTI.

### Conclusions

HBOT is safe and beneficial for treating chronic wounds and LRTI in the sarcoma population. Awaiting further prospective results, we recommend referring to HBOT centers more actively in patients experiencing impaired wound healing or symptoms of delayed radiation-induced tissue injury following multimodality sarcoma treatment.

   

**Data Availability Statement:** The data cannot be made publicly available due to ethical restrictions as data were pseudo-anonimized by using an

identification code, but place of treatment and several dates related to the patient could make it identifiable. The data underlying the results presented in the study are available from J.T Bottema at j.t.bottema@umcg.nl.

**Funding:** The author(s) received no specific funding for this work.

**Competing interests:** The authors have declared no competing interests exist.

## Introduction

Sarcomas are malignant neoplasms originating from mesenchymal tissues, broadly divided by histology in soft-tissue and bone sarcomas, responsible for approximately 1% and 0.2% of all malignancies in adults, respectively [1]. The incidence of bone and soft-tissue sarcomas together is 900–1000 cases in the Netherlands per year [2].

Sarcoma management is complex and usually involves multiple modalities, such as radiotherapy (RT) and in some cases chemotherapy (CT). Surgical excision is the cornerstone of treatment and is essential to achieve local tumor control [3,4]. RT can be required to minimize local recurrence [5], but is frequently complicated by radiation-induced tissue injury [6–8]. Acute injury presents during or immediately following RT and is generally self-limiting. Late radiation tissue injury (LRTI) is characterized by a symptom-free latent period of three to six months or more, after which complications such as subcutaneous tissue fibrosis, edema, joint stiffness, osteoradionecrosis and fracture may occur [9,10].

The RT of choice in sarcoma management is external-beam radiotherapy (EBRT) and is administered in either the preoperative or postoperative setting. Both settings cause radiation-induced injury [8,11], but the timing influences the pattern and type of injury. The postoperative major wound complication rate is approximately 35% for preoperative EBRT, while postoperative EBRT leads to wound complications in 17% of patients [6–8]. Delayed wound healing, among others, is classified as a major wound complication [7]. Furthermore, high rates of LRTI have been reported in both preoperative and postoperative EBRT. Postoperative EBRT, which is characterized by larger radiation fields and higher radiation doses, more often results in LRTI [11–13]. Fibrosis occurs in around 48% of postoperatively irradiated patients and 32% of preoperatively irradiated patients, joint stiffness in 23% and 18% and edema in 23% and 15%, respectively [10,11,14]. Complications following sarcoma treatment result in detrimental functional outcome and general health status of patients [12,13].

Hyperbaric oxygen therapy (HBOT) is a therapy that comprises the administration of 100% oxygen at an elevated environmental pressure, which results in a direct increase in tissue oxygenation [15]. Also, HBOT mobilizes stem cells that enhance vasculogenesis [16,17]. These mechanisms result in improved tissue quality by neovascularization [9,18,19] and have shown to be beneficial in radiation proctitis and LRTI involving bone and soft tissues of the head and neck [9,20,21]. Furthermore, the application of HBOT after combined surgery and RT in head and neck tumors has shown to lead to improved healing of chronic wounds [22].

This retrospective cohort study aims to provide an overview of the use and results of HBOT for radiation-induced tissue injuries following multimodality sarcoma treatment. It is hypothesized that HBOT improves wound healing and LRTI for radiation-induced tissue injury in sarcoma patients.

## Materials and methods

### Patients

The Institutional Review Board of the University Medical Center Groningen (UMCG) approved this retrospective study (case number 2017.521). Informed consent was obtained from all patients included in the study. All sarcoma patients who were treated between January 2006 and October 2017 in one of the five centers of the Institute for Hyperbaric Medicine in the Netherlands were included. Patients were referred to the HBOT centers by medical specialists from various medical centers in the Netherlands for presence of chronic wounds or LRTI. Wounds were classified as chronic when not healed within three months. LRTI was defined as symptomatic tissue injury due to radiation toxicity after three months or more following

completion of RT. The effects of HBOT were determined by either the patient or the doctor during follow up. Patients' medical records were analyzed by a medical doctor to gain insight in patient and tumor characteristics, sarcoma treatment, indications for HBOT and HBOT results.

## HBOT procedure

Patients underwent daily HBOT sessions (excluding weekends) for six to eight weeks in a hyperbaric chamber in one of the five facilities of the Institute of Hyperbaric Medicine in the Netherlands. In the chamber, the pressure was elevated to 2.5 atmospheres absolute (ATA) in ten minutes. When pressure reached 2.5 ATA, 100% oxygen was administrated in three sessions of 20 minutes by face mask, alternated with five minutes of room air, followed by a final 15 minutes of oxygen therapy. After this, pressure was lowered in ten minutes, with decompression on oxygen up to 1.3 ATA. This results in a total treatment duration of 110 minutes, with patients being exposed to hyperbaric oxygen at 2.5 ATA for a total of 90 minutes [21].

## Statistical analysis

Quantitative analyses were performed using SPSS® Version 23.0 (IBM SPSS Statistics for Windows, Version 23.0 Armonk, NY: IBM Corp). Categorical variables were described using frequencies and percentages, numerical variables by median and interquartile ranges (IQR) or by mean and standard deviation (SD) as appropriate.

# Results

## Population

In total, 30 patients were included in this study, 18 men (60.0%) and 12 women (40.0%). The age of patients at initiation of HBOT ranged from 19 to 86 years and median age was 67.0 (55.3–74.5) years. The distribution of patients among different HBOT centers was as follows: 13 (43.3%) patients were treated in center A, seven (23.3%) in center B, five (16.7%) in center C, three (10.0%) in center D and two (6.7%) in center E. Time between surgery and referral was median 9 (2.8–26.8) months.

More than half of the patients, 16 (53.3%), had no history of smoking. Six (20.0%) patients had stopped smoking more than six months prior to therapy and two (6.7%) patients stopped smoking within six months prior to the start of HBOT. There were five (16.7%) patients still smoking at HBOT initiation. Diabetes mellitus (DM) was present in four (13.3%) patients and absent in 25 (83.3%) patients. Distinction between sarcoma groups was based on pathology results from the medical center from which the patient was referred. Type of sarcoma was bone in four (13.3%) and soft-tissue in 26 (86.7%) patients. Sarcomas were located in the trunk/thorax in seven (23.3%) patients, upper extremity in six (20.0%) patients, lower extremity in 16 (53.3%) and head/neck in one (3.3%) patient.

Sarcoma treatment consisted of only RT in one (3.3%) patient, RT and CT in one (3.3%) patient, RT, CT and resection in two (6.7%) patients and RT and resection in 26 (86.7%) patients. Wound closure after sarcoma resection was primary in 22 (73.3%), split skin graft (SSG) in one (3.3%) patient and by application of a skin flap in four (13.3%) patients. All patients were treated with EBRT. EBRT was administered in the preoperative setting in seven (23.3%) patients and postoperative in 22 (73.3%) patients. Preoperative patients were irradiated with a total dose of 50 Gy in 2 Gy fractions. The postoperative patients received a median total dose of 66.3 (60.0–70.0) Gy and all but two patients were treated in fractions of 2 Gy. These two patients were irradiated in fractions of 1.9 and 2.5 Gy. The patient that was treated

by RT and CT received a total dose of 45.0 Gy in fractions of 1.8 Gy. Table 1 presents patient, tumor and treatment characteristics.

## HBOT characteristics

Indications for HBOT were chronic wounds in 18 (60.0%) patients and LRTI in 12 (40.0%) patients. Of the 30 patients, 23 (76.7%) finished all sessions according to the treatment plan. Two patients ended HBOT because of diagnosis with lung metastases. Furthermore, presence of a lung embolus, treatment takeover by an academic center, claustrophobia, suspicion of a leg tumor and viral infection were reasons to stop for the other patients. The patients that did not continue HBOT because of claustrophobia and takeover by an academic center only underwent one and four sessions, respectively. These patients were therefore excluded from analysis of HBOT effects. The median of HBOT sessions was 40 (30–40) for the 28 remaining patients.

## Adverse events

Twenty-two (73.3%) patients reported no adverse events, the other eight (26.7%) patients did. Barotrauma to the middle ear due to differences in ambient pressure was observed via oto-scopy in five (55.5%) patients. Pressure equalization (PE) tubes were inserted in three of five patients. In one of the five patients, tympanocentesis for otitis media with effusion and vertigo was performed, which relieved the complaints. Myopia was reported two (22.2%) times. Fatigue was recorded one (11.1%) time. The patient that ended HBOT after one session reported claustrophobia (11.1%) as adverse event. One of the eight patients that experienced adverse events, had both myopia and ear problems for which a PE tube was placed.

## Effects on chronic wounds and LRTI

Fifteen of the 16 chronic wounds were located at the site of the surgical wound. One wound spontaneously occurred in irradiated area near to the operation site. Of the 16 patients that received HBOT for chronic wounds, ten had data available on wound measurements before and after treatment. Improved wound healing was seen in all of these ten patients. In total, in 11 of 16 (68.8%) patients a trend towards healing of the wound was seen. Of these 11 patients that showed improved wound healing, complete wound healing, either (delayed) primary or secondary, was observed in seven patients. This corresponds with 43.8% of the total number of patients treated for chronic wounds. Table 2 presents effects of HBOT on chronic wounds. Fig 1 shows pictures of a patient referred from our medical center for a chronic wound in the upper leg before and after HBOT.

Twelve patients were referred to HBOT centers for LRTI, including pain, fibrosis, osteora-dionecrosis and radiation cystitis. Six of these patients were referred because of pain complaints. All of these patients reported a decrease in pain. In total, nine out of 12 (75.0%) patients experienced a decline in pain. Also, eight patients underwent HBOT for fibrosis and in five of eight (62.5%) patients an improvement of fibrosis was reported. Increase of joint function was reported twice. HBOT was indicated for osteoradionecrosis in one patient, but no objective results were reported. One patient received HBOT for hematuria resulting from radiation cystitis and hematuria resolved completely. Results of HBOT on indication for referral and other symptoms of LRTI are presented in Table 3.

**Table 1. Patient, tumor and sarcoma treatment characteristics.**

| Characteristic | N = 30 |
|---|---|
| **Gender** | |
| Male | 18 (60.0%) |
| Female | 12 (40.0%) |
| **Age (years)** | 67.0 (55.3–74.5) |
| **HBOT center** | |
| Center A | 13 (43.3%) |
| Center B | 7 (23.3%) |
| Center C | 5 (16.7%) |
| Center D | 3 (10.0%) |
| Center E | 2 (6.7%) |
| **Time between surgery and referral (months)** | 12.5 (2.8–26.8) |
| **Smoking** | |
| Never | 16 (53.3%) |
| Stopped since > 6 months | 6 (20.0%) |
| Stopped since < 6 months | 2 (6.7%) |
| Yes | 5 (16.7%) |
| Missing | 1 (3.3%) |
| **DM** | |
| Yes | 4 (13.3%) |
| No | 25 (83.3%) |
| Missing | 1 (3.3%) |
| **Sarcoma type** | |
| Bone | 4 (13.3%) |
| Soft-tissue | 26 (86.7%) |
| **Sarcoma location** | |
| Trunk/thorax | 7 (23.3%) |
| Upper extremity | 6 (20.0%) |
| Lower extremity | 16 (53.3%) |
| Head/neck | 1 (3.3%) |
| **Sarcoma treatment** | |
| RT | 1 (3.3%) |
| RT + CT | 1 (3.3%) |
| RT + CT + resection | 2 (6.7%) |
| RT + resection | 26 (86.7%) |
| **Wound closure** | |
| Primary | 22 (73.3%) |
| Skin flap | 4 (13.3%) |
| SSG | 1 (3.3%) |
| Missing | 3 (10.0%) |
| **Timing of EBRT** | |
| Preoperative | 7 (23.3%) |
| Postoperative | 22 (73.3%) |
| Missing | 1 (3.3%) |
| **Total dosage of RT administered (Gy)** | |
| Preoperative | 50 (50–50) |
| Postoperative | 66.3 (60–70) |

Data is presented as n (%), mean (SD) or median (IQR). Abbreviations. HBOT: hyperbaric oxygen therapy. DM: Diabetes Mellitus. EBRT: external-beam radiotherapy. SSG: split skin graft. RT: radiotherapy. CT: chemotherapy.

**Table 2. Effects of HBOT on chronic wounds.**

| Patient | Sex | Age (years) | Smoking | DM | Timing of RT | Total dose of RT (Gy) | Number of HBOT sessions | Wound size prior to HBOT | Wound size after HBOT | Results |
|---|---|---|---|---|---|---|---|---|---|---|
| 1 | Female | 21 | No, never | No | Postoperative | Not recorded | 20 + 10 | Not recorded | 3x4 cm | Wound closure by local mesoplastics. |
| 2 | Male | 69 | No, never | No | Postoperative | 70 | 30 | 15.8x10.4 cm | Not recorded | Improved wound healing at 20 HBOT sessions. |
| 3 | Male | 61 | Yes | No | Postoperative | 60 | 30 + 10 | Not recorded | Not recorded | Cleaner, granulating wound, closure of the wound by free flap. |
| 4 | Male | 84 | No, never | No | Postoperative | 50 | 40 | 3 cm deep | 2.9 cm deep | Granulating wound. |
| 5 | Male | 76 | No, since > 6 months | No | Postoperative | 64 | 45 + 20 | 14x14 cm | 10x7 cm | Improved wound healing, wound closure by SSG. |
| 6 | Male | 73 | Yes | No | Postoperative | 70 | 40 | 2x1 cm | | Complete healing. |
| 7 | Female | 68 | No | No | Preoperative | 50 | 30 | 2x3x10 cm | 1x1 cm (4–5 cm beneath the skin) | Improved wound healing. |
| 8 | Male | 70 | No | No | Preoperative | 50 | 30 | 6x2x1 cm | | Complete healing. |
| 9 | Female | 69 | No | No | Postoperative | 60 | 40 | 8 cm deep | Not recorded | Closure by tensor fascia lata free flap after 40 sessions. |
| 10 | Male | 82 | No | No | Preoperative | 50 | 35 | 12x8x2 cm | 5 cm in length | Superficial defect, improved wound healing. |
| 11 | Male | 74 | No | No | Preoperative | 50 | 30 | 8x1x0.4 cm | | Complete healing. |
| 12 | Male | 86 | No, since < 6 months | No | Postoperative | 70 | 40 | 4x5 cm | 3x4 cm | Improved wound healing, more granulation tissue. |
| 13* | Female | 69 | No | Yes | Postoperative | Not recorded | 26 | 5x2.5x3 cm | Not recorded | Cleaner wound, need for surgical closure (but impossible due to cardiac problems). |
| 14 | Female | 63 | No | Yes | Preoperative | 50 | 27 | 8 cm deep | 5 cm deep | Improved wound healing, clean wound. |
| 15 | Male | 78 | Yes | No | Preoperative | 50 | 33 | 1 cm | Not recorded | Improved wound healing (wound almost closed), reduction of drain fluid. |
| 16 | Female | 73 | No, since < 6 months | Yes | Postoperative | 60 | 36 | 2.5x0.2 cm | 2x0.2 cm | Improved wound healing. |

Abbreviations. DM: Diabetes Mellitus. RT: radiotherapy. HBOT: hyperbaric oxygen therapy. SSG: split skin graft. *This patient had a chronic wound near to the operation site in irradiated area. All other patients had non-healing surgical wounds.

## Discussion

No recent literature is available on HBOT outcome in chronic wounds and delayed radiation injury in the sarcoma population. In a case series of 17 patients receiving HBOT for chronic necrotic wounds in extremities after RT for a variety of malignancies, including eight soft-tissue sarcomas, Feldmeier et al. (2000) reported complete wound healing in 85% of patients when also taking into account delayed primary closed wounds [23]. In our cohort, complete wound healing was seen in seven of 16 (43.8%) patients. The difference in rates may result from dissimilar population characteristics. HBOT is more accepted as an adjunct to treatment in a variety of other malignancies treated with RT, both originating from soft-tissue and bone, such as head/neck tumors and tumors in the pelvic region [9,18,20,24,25]. In these studies, HBOT results in improvement of chronic wounds and LRTI.

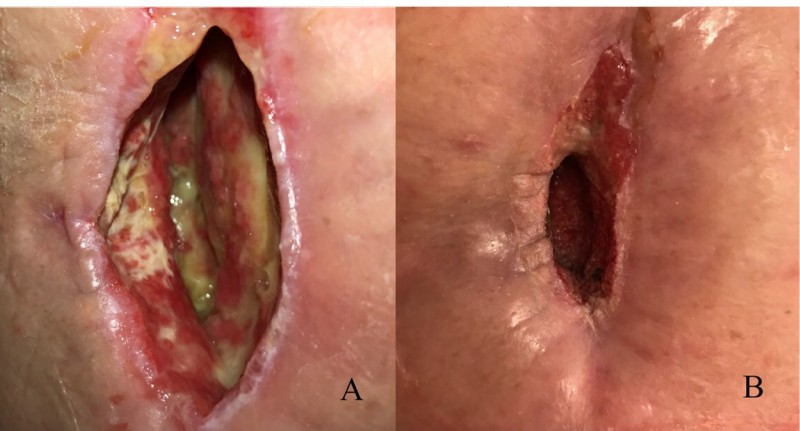

**Fig 1. Pictures of a patient referred from our medical center for a chronic wound of the medial upper leg after limb-salvage therapy, consisting of surgery and preoperative radiotherapy.** A, At HBOT initiation, 11 weeks after completion of limb-salvage therapy. B, Improved wound healing is seen after 40 sessions of HBOT.

In our medical center, HBOT is offered to a selection of patients suffering from radiation-induced tissue injuries in a late stadium and it is considered as a 'last resort' treatment option. Since median time between surgery and referral for HBOT was 9 months in the cohort, this view on HBOT is thought to be shared by specialists in the Netherlands. Of note, the onset of chronic wounds is frequently in the first few weeks after primary closure. In general, these wounds show inferior healing tendency rapidly after surgery and open shortly after sutures are removed. The restrained attitude towards HBOT is due to lack of evidence for efficacy of the therapy in sarcoma patients in our medical center. As a consequence, the patients referred to receive HBOT are usually the ones with the most severe complications. This results in a study population not representative of the actual patient population.

This retrospective cohort analysis has limitations resulting from the nature of the study. Data from patient records were gathered retrospectively, which leads to missing data, subjective outcomes and a limited population size. Outcomes are partly subjective, because in 10/16 patients treated for chronic wounds, wound measurements by a medical doctor before and after hyperbaric oxygen therapy are reported. Also, all sarcoma patients treated in over ten years in five centers in the Netherlands have been included for analysis. Another five centers facilitate HBOT in the Netherlands, however the Institute of Hyperbaric Medicine is the largest and its centers are widely dispersed over the Netherlands.

Radiation-induced tissue injury originates from either direct or indirect molecular effects of ionization, of which indirect effects result from emergence of radicals. Since completely intact DNA is needed for cell proliferation, damage to DNA and insufficient repair of DNA are frequent causes of cell death in radiated tissues. Therefore, cell survival in radiated tissue is based on cell proliferation frequency. Acute radiation injury is seen in tissues with high mitotic activity, such as tumors, but also in intestinal mucosa, the skin and hematopoietic stem cells [26]. Damage to blood vessels is a delayed type of radiation injury, occurs after several months and plays an important role in pathophysiology of chronic wounds and LRTI because of the creation of an hypoxic, hypocellular and hypovascular environment: "the 3H's" [27,28]. Furthermore, release of cytokines, subsequent fibroblast proliferation and fibro-atrophic processes are hypothesized to form the basis of LRTI [29].

Marx et al. demonstrated that by administering oxygen at high pressure, vascularity and cellularity in tissues is increased [19]. Besides, HBOT has multiple other effects on tissue, such as

**Table 3. Effects of HBOT on LRTI.**

| Patient | Sex | Age (years) | Smoking | DM | Timing of RT | Total dose of RT (Gy) | Number of HBOT sessions | LRTI indication | Results on LRTI indication | Other results |
|---|---|---|---|---|---|---|---|---|---|---|
| 17 | Male | 77 | No, since > 6 months | No | Postoperative | 70 | 30 | Radiation-induced injury of soft tissues | Decrease of pain, edema and fibrosis. | |
| 18 | Female | 47 | No, since > 6 months | No | Postoperative | 70 | 40 | Pain, fibrosis | Less pain. | Decrease in edema, erythema and increase in function of arm. |
| 19 | Female | 62 | Yes | No | Postoperative | 66.5 | 40 | Pain, fibrosis | Less pain, but pain medication was also altered during HBOT. | No change in joint function. |
| 20 | Female | 66 | No | No | Postoperative | 70 | 40 | Pain, fibrosis | Decrease in pain and fibrosis. | Increase in knee joint function. |
| 21 | Female | 47 | No, since > 6 months | No | Postoperative | 66 | 40 | Pain, fibrosis | Decrease in pain and fibrosis. | |
| 22 | Male | 64 | No, since > 6 months | No | Postoperative | 70 | 40 | Pain, fibrosis | Decrease in pain and fibrosis. | |
| 23 | Female | 85 | No | No | Postoperative | 60 | 29 | Pain | Slight improvement of pain. | HBOT was seen as intensive. |
| 24 | Female | 65 | Yes | No | Postoperative | 70 | 40 | Osteoradionecrosis | No objective results regarding to the osteoradionecrosis. | Less pain, decreased use of pain medication, improvement of defecation pattern. |
| 25 | Male | 44 | Not recorded | Not recorded | Postoperative | 66 | 28 + 10 | Fibrosis | Decrease in fibrosis. | |
| 26 | Male | 19 | No | No | No operation | 45 | 40 | Radiation cystitis | Resolution of hematuria. | |
| 27 | Male | 61 | No | No | Postoperative | 70 | 40 | Fibrosis | Decrease in fibrosis. | Decrease in pain. |
| 28 | Male | 58 | No, since > 6 months | Yes | Postoperative | 60 | 40 | Fibrosis | | Decrease in pain. Less problems with defecation and micturition. |

Abbreviations. DM: Diabetes Mellitus. RT: radiotherapy. HBOT: hyperbaric oxygen therapy. LRTI: late radiation tissue injury.

improvement of white cell and fibroblast function and mobilization of stem cells [9,17,18]. These physiologic changes are hypothesized to result in the beneficial consequences of HBOT administration. There have been concerns about HBOT as being carcinogenic by promoting cell proliferation, but published literature does not support this [30]. Also, some adverse events are seen in HBOT, such as damage to the ears and temporary worsening of myopia [9]. In our population, HBOT was generally well tolerated (73.3% reported no adverse events) and complications were either treatable or not severe.

As survival rates for sarcomas ameliorate [31] and timing of RT shifts to being preoperatively administered [5], interest increases in finding new treatment options for radiation-induced tissue injuries. Preoperative RT leads to higher rates of postoperative wound complications (35% versus 17% for postoperative RT) [6–8] due to performing surgical intervention in already ischemic tissue, as a consequence of RT [29,32]. However, Davis et al. report similar results in functional outcome when comparing preoperative to postoperative RT at two years after treatment. Little is known about consequences of the shift of RT timing past those two

years. Since wound complications, such as chronic wounds, are seen more frequently in the preoperatively irradiated population and LRTI is seen even after several years following RT, knowledge on the utilization of therapeutic options to reduce complications after sarcoma management continues to be relevant in the future. HBOT outcome on wound healing following preoperative RT and surgical intervention in sarcoma patients is currently being evaluated in a prospective, randomized study (NCT03144206) [33].

## Conclusions

The aim of this retrospective analysis of a Dutch cohort is to gain insight in the use and results of HBOT for radiation-induced tissue injuries following multimodality sarcoma management. The patients referred to HBOT centers were the ones suffering from severe complications in a late stage after sarcoma treatment. HBOT is regarded as safe and beneficial for treating both chronic wounds and LRTI in these cases. Awaiting further prospective results, we recommend referring to HBOT centers more actively in an earlier stage in sarcoma patients experiencing impaired wound healing or symptoms of delayed radiation-induced tissue injury following multimodality sarcoma treatment.

## Author Contributions

**Conceptualization:** Barbara L. van Leeuwen, Marc G. Stevenson, Lukas B. Been.

**Data curation:** Corine A. Lansdorp, Onno Boonstra, Marc G. Stevenson.

**Formal analysis:** Jasmijn D. Generaal.

**Project administration:** Marc G. Stevenson.

**Resources:** Corine A. Lansdorp, Onno Boonstra.

**Supervision:** Barbara L. van Leeuwen, Marc G. Stevenson, Lukas B. Been.

**Visualization:** Jasmijn D. Generaal, Marc G. Stevenson.

**Writing – original draft:** Jasmijn D. Generaal.

**Writing – review & editing:** Corine A. Lansdorp, Onno Boonstra, Barbara L. van Leeuwen, Hubertus A. M. Vanhauten, Marc G. Stevenson, Lukas B. Been.

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
