## [Decision Letter · Decision Letter 0]

27 Nov 2019

PONE-D-19-28754

Hyperbaric oxygen therapy for radiation-induced tissue injury following sarcoma treatment: a retrospective analysis of a Dutch cohort.

PLOS ONE

Dear Dr. Been,

Thank you for submitting your manuscript to PLOS ONE. After careful consideration, we feel that it has merit but does not fully meet PLOS ONE’s publication criteria as it currently stands. Therefore, we invite you to submit a revised version of the manuscript that addresses the points raised during the review process.

We would appreciate receiving your revised manuscript by Jan 11 2020 11:59PM. To enhance the reproducibility of your results, we recommend that if applicable you deposit your laboratory protocols in protocols.io, where a protocol can be assigned its own identifier (DOI) such that it can be cited independently in the future. For instructions see: http://journals.plos.org/plosone/s/submission-guidelines#loc-laboratory-protocols

We look forward to receiving your revised manuscript.

Kind regards,

Brian E. Brigman

Academic Editor

PLOS ONE

Journal Requirements:

2. In ethics statement in the manuscript and in the online submission form, please provide additional information about the patient records used in your retrospective study. Specifically, please ensure that you have discussed whether all data were fully anonymized before you accessed them and/or whether the IRB or ethics committee waived the requirement for informed consent. If patients provided informed written consent to have data from their medical records used in research, please include this information.

Additional Editor Comments:

This is a retrospective review of 30 patients with chronic radiation injury or chronic wounds after radiation for sarcoma. This report suffers the usual issues with retrospective reviews. In addition, the authors need to address the following points:

No clear hypothesis.

No control group makes this data difficult to interpret.

There are subjective outcomes with significant amounts of data missing.

No justification of sample size.

Please note the scope of the journal: https://journals.plos.org/plosone/s/journal-information#loc-scope

Reviewers' comments:

Reviewer's Responses to Questions

**Comments to the Author**

1. Is the manuscript technically sound, and do the data support the conclusions?

Reviewer #1: Yes

2. Has the statistical analysis been performed appropriately and rigorously? 

Reviewer #1: Yes

3. Have the authors made all data underlying the findings in their manuscript fully available?

Reviewer #1: Yes

4. Is the manuscript presented in an intelligible fashion and written in standard English?

Reviewer #1: Yes

5. Review Comments to the Author

Reviewer #1: Congratulations to the authors on a well-written, informative, and important manuscript. HBOT is a modality which we often discuss and occasionally employ at my institution. Although I would ultimately like to know the role for HBOT in the setting of NA RT *prior to* the onset of complications, this work provides an important first step.

I have a few small requests of the authors:

1) Lines 71-74, I would like to see separate references for the two claims - direct increase in tissue oxygenation (which is very well described) and release of bone marrow stem cells (which is a slightly more controversial topic). I do not agree that these two mechanisms should be lumped together in a single sentence with a single reference to support.

2) Lines 165-166. I was confused by the sentence "No barotrauma to the tympanic membrane was observed." It seems that if 55% of the patients had "ear problems" and 3 of those 5 patients required PE tube insertion then it is unlikely that there was any barotrauma to the TM. Can the authors please use more specific language to replace "ear problems"? And can they please clarify how they defined barotrauma? Perhaps they meant that 5 patients had symptoms related to equalization of pressure within the ear but there was no TM rupture. In my experience, most patients with symptoms referable to the ear have changes in the exam of the TM consistent with barotrauma, even if a rupture has not occurred. Clarification would be helfpul to the reader.

Aside from these small comments, the overall clarity of the authors' work - especially the Conclusions - adds to the value of this paper.

6. PLOS authors have the option to publish the peer review history of their article (what does this mean?). If published, this will include your full peer review and any attached files.

Reviewer #1: Yes: William Eward

---

## [Author Response · Author response to Decision Letter 0]

4 Feb 2020

Thank you for the review of our manuscript “Hyperbaric oxygen therapy for radiation-induced tissue injury following sarcoma treatment: a retrospective analysis of a Dutch cohort.”. We have changed the manuscript according to the feedback and will discuss our considerations in detail in the following paragraphs.

The first request from the reviewer to the authors is to provide separate references for two claims that are made in lines 71-74 in the original manuscript, in which the mechanisms of hyperbaric oxygen therapy are explained. The first claim includes the direct increase of tissue oxygenation after treatment with hyperbaric oxygen therapy. The second claim involves the induction and mobilization of stem cells. Both mechanisms are thought to be at the base of the angiogenesis by hyperbaric oxygen therapy. For the first claim, the work of Hills (1999) is used, in which the physiologic mechanism behind the direct increase of tissue oxygenation is explained. For the second claim, the papers of Milovanova et al. (2009) and Thom et al. (2006) are used as a reference, in which the stem cell mobilization and pathway of vasculogenesis are researched by using murine models. Furthermore, to state the effects of hyperbaric oxygen therapy on angiogenesis, we refer to the paper of Marx et. al (1990) and two reviews; Feldmeier (2012) and the Cochrane review by Bennett et al (2016). Marx et al. (1990) demonstrates the increased vascular density after hyperbaric administration of oxygen, instead of normobaric oxygen. 

The second request is to clarify the complications of hyperbaric oxygen therapy related to the ears. By requesting additional information from the researchers in the Institutes of Hyperbaric Medicine, we have improved this section. Fifty-five percent of patients that experienced adverse events had barotrauma to the middle ear due to changes in pressure. This was observed by the doctor via otoscopy.

Additional comments on the manuscript were made by the editor. These include the absence of a clear hypothesis, the absence of a control group, missing data, subjective outcomes and no justification of sample size. In the introduction of the revised manuscript, a more specific hypothesis has been added. Other comments are addressed in a new paragraph in the discussion. It is explained that this retrospective cohort study has limitations resulting from the nature of the study; the results should of course be interpreted bearing this in mind. We noticed the gap of knowledge on effects of hyperbaric oxygen therapy and the clinical relevance of finding treatment options for radiation-induced tissue injury in the sarcoma population. The use of hyperbaric oxygen therapy remains subject of debate in our medical center, which is also stated by the reviewer. We retrospectively analyzed patient records of patients that were referred to different centers of the Institute of Hyperbaric Oxygen as a ‘last resort’ option. Outcomes are partly subjective and there is no control group as data were gathered retrospectively. Of note, in 10/16 patients referred for chronic wounds, wound measurements by a medical doctor were reported before and after hyperbaric oxygen therapy (Table 2). In all of these patients, the effect of hyperbaric oxygen therapy can be objectivized. There was no sample size calculation, because our population size is completely dependent on the amount of referrals to the Institute of Hyperbaric Oxygen in more than a ten year period. We analyzed data of all sarcoma patients treated in these centers. We strongly believe that this manuscript is a first step in making hyperbaric oxygen therapy more accepted as an addition to the multidisciplinary treatment of sarcoma patients. It provides a clear overview of the effects and adverse events of hyperbaric oxygen therapy and it seems to be a safe and beneficial treatment option for this patient population that struggles with treatment-induced morbidity. The paper offers a new option to decrease the treatment-induced morbidity and incites new, prospective, research. 

Further journal requirements involve style requirements, the ethical statement and data management. The style of the references has been adjusted to the templates provided. The ethical statement in the manuscript is elaborated by adding information on informed consent. Data were pseudo-anonimized by using an identification code, but place of treatment and several dates related to the patient could make it identifiable. So, the data cannot be made publicly available due to ethical restrictions. However, the result section includes two tables with anonimized patient data, to enable readers to objectively judge the rationale behind our conclusions. Requests on data can be sent to the corresponding author. 

With this letter and the adjustments made in the manuscript, we hope to have adequately answered the questions and comments. We would be grateful if you would take this manuscript into consideration for publication.

---

## [Decision Letter · Decision Letter 1]

2 Apr 2020

PONE-D-19-28754R1

Hyperbaric oxygen therapy for radiation-induced tissue injury following sarcoma treatment: a retrospective analysis of a Dutch cohort.

PLOS ONE

Dear Dr. Been,

Thank you for submitting your manuscript to PLOS ONE. After careful consideration, we feel that it has merit but does not fully meet PLOS ONE’s publication criteria as it currently stands. Therefore, we invite you to submit a revised version of the manuscript that addresses the points raised during the review process.

We would appreciate receiving your revised manuscript by May 17 2020 11:59PM. To enhance the reproducibility of your results, we recommend that if applicable you deposit your laboratory protocols in protocols.io, where a protocol can be assigned its own identifier (DOI) such that it can be cited independently in the future. For instructions see: http://journals.plos.org/plosone/s/submission-guidelines#loc-laboratory-protocols

We look forward to receiving your revised manuscript.

Kind regards,

Brian E. Brigman

Academic Editor

PLOS ONE

Additional Editor Comments (if provided):

Please include the data requested by reviewer 2.

Reviewers' comments:

Reviewer's Responses to Questions

**Comments to the Author**

1. If the authors have adequately addressed your comments raised in a previous round of review and you feel that this manuscript is now acceptable for publication, you may indicate that here to bypass the “Comments to the Author” section, enter your conflict of interest statement in the “Confidential to Editor” section, and submit your "Accept" recommendation.

Reviewer #1: All comments have been addressed

Reviewer #2: (No Response)

2. Is the manuscript technically sound, and do the data support the conclusions?

Reviewer #1: Yes

Reviewer #2: Yes

3. Has the statistical analysis been performed appropriately and rigorously? 

Reviewer #1: Yes

Reviewer #2: N/A

4. Have the authors made all data underlying the findings in their manuscript fully available?

Reviewer #1: Yes

Reviewer #2: No

5. Is the manuscript presented in an intelligible fashion and written in standard English?

Reviewer #1: Yes

Reviewer #2: Yes

6. Review Comments to the Author

Reviewer #1: The authors should be congratulated on a well-revised manuscript which illustrates the possible utility in using HBOT to treat LRTI. This is a significant problem that causes great morbidity in patients treated with multimodal therapy for soft tissue sarcoma. This work lays the foundation for subsequent prospective studies.

Reviewer #2: This is a useful case series describing a group of patients treated with hyperbaric oxygen for wounds related to radiation injury. The only data needed is related to the onset of the wounds. It is unclear whether these wounds were present since surgery, or occurred some time afterward. A column should be added to Table 2 to provide the time of occurrence of the wound. In addition, the authors should state whether the wounds were at the site of surgery or somewhere else.

7. PLOS authors have the option to publish the peer review history of their article (what does this mean?). If published, this will include your full peer review and any attached files.

Reviewer #1: Yes: William Eward

Reviewer #2: No

---

## [Author Response · Author response to Decision Letter 1]

29 Apr 2020

Dear dr. B.E. Brigman and reviewers,

Thank you for the review of our manuscript “Hyperbaric oxygen therapy for radiation-induced tissue injury following sarcoma treatment: a retrospective analysis of a Dutch cohort.”. We have made adjustments to the manuscript according to the feedback provided.

Reviewer 2 commented the following: “This is a useful case series describing a group of patients treated with hyperbaric oxygen for wounds related to radiation injury. The only data needed is related to the onset of the wounds. It is unclear whether these wounds were present since surgery, or occurred some time afterward. A column should be added to Table 2 to provide the time of occurrence of the wound. In addition, the authors should state whether the wounds were at the site of surgery or somewhere else.” In 15 of 16 patients referred for chronic wounds, hyperbaric oxygen therapy was indicated for a chronic wound at the site of surgery. The remaining patient suffered from a wound that spontaneously occurred near to the operation site (in irradiated area). Adjustments have been made to the results-section and table 2 to clarify the site of chronic wounds. 

Furthermore, time between surgery and referral to an Institute for Hyperbaric Medicine was less than 12 months for 14 of 16 patients suffering from chronic wounds. All of these 14 patients had a chronic wound at the site of surgery. In clinical practice, inferior healing tendency is seen rapidly after surgery, frequently soon after sutures are removed. Time to referral was more than 12 months for the other two patients. One of these patients was referred for a persistent wound more than two years after surgery and radiotherapy. The remaining patient is the one that was referred for a wound in irradiated area, in which the onset is unknown. The patient was referred more than four years after surgery. 

During the review of patient files to provide more detailed data on location and onset of the chronic wounds, it was found out that date of surgery was incorrect for one patient in the dataset. The median time to referral was changed accordingly in the manuscript.

With this letter and the adjustments made in the manuscript, we hope to have adequately answered the questions and comments. 

Yours sincerely on behalf of all authors,

L.B. Been, MD PhD

University of Groningen, University Medical Center Groningen

Department of Surgery, division of Surgical Oncology BA31

PO Box 30.001

9700 RB Groningen, The Netherlands

Phone: +31-503612317 ; Fax: +31-503611745

Email: l.b.been@umcg.nl

---

## [Decision Letter · Decision Letter 2]

27 May 2020

Hyperbaric oxygen therapy for radiation-induced tissue injury following sarcoma treatment: a retrospective analysis of a Dutch cohort.

PONE-D-19-28754R2

Dear Dr. Been,

We are pleased to inform you that your manuscript has been judged scientifically suitable for publication and will be formally accepted for publication once it complies with all outstanding technical requirements.

With kind regards,

Brian E. Brigman

Academic Editor

PLOS ONE

Additional Editor Comments (optional):

Reviewers' comments:

Reviewer's Responses to Questions

**Comments to the Author**

1. If the authors have adequately addressed your comments raised in a previous round of review and you feel that this manuscript is now acceptable for publication, you may indicate that here to bypass the “Comments to the Author” section, enter your conflict of interest statement in the “Confidential to Editor” section, and submit your "Accept" recommendation.

Reviewer #1: All comments have been addressed

Reviewer #2: All comments have been addressed

2. Is the manuscript technically sound, and do the data support the conclusions?

Reviewer #1: Yes

Reviewer #2: Yes

3. Has the statistical analysis been performed appropriately and rigorously? 

Reviewer #1: Yes

Reviewer #2: N/A

4. Have the authors made all data underlying the findings in their manuscript fully available?

Reviewer #1: Yes

Reviewer #2: Yes

5. Is the manuscript presented in an intelligible fashion and written in standard English?

Reviewer #1: Yes

Reviewer #2: Yes

6. Review Comments to the Author

Reviewer #1: I appreciate the efforts the authors have made to address my concerns and those of the other reviewers. The manuscript will be valuable to multiple different segments of the medical community.

Reviewer #2: (No Response)

7. PLOS authors have the option to publish the peer review history of their article (what does this mean?). If published, this will include your full peer review and any attached files.

Reviewer #1: No

Reviewer #2: Yes: Richard Moon, MD

---

## [Editor Report · Acceptance letter]

29 May 2020

PONE-D-19-28754R2 

Hyperbaric oxygen therapy for radiation-induced tissue injury following sarcoma treatment: a retrospective analysis of a Dutch cohort. 

Dear Dr. Been:

I am pleased to inform you that your manuscript has been deemed suitable for publication in PLOS ONE. Congratulations! Your manuscript is now with our production department. 

With kind regards,

on behalf of

Dr. Brian E. Brigman 

Academic Editor

PLOS ONE